Susceptibility of flexible plastic foodstuffs packaging against Monomorium indicum (Hymenoptera: Formicidae) household ants

Iqbal Muhammad Junaid 1
Hassan Muhammad Waqar waqar.hassan@iub.edu.pk 1
Sarwar Ghulam 2
Jamil Moazzam 3
Hussain Tanveer 4
1 Department of Entomology, Faculty of Agriculture and Environment, Islamia University of Bahawalpur , Bahawalpur , Punjab , Pakistan
2 Department of Biology, Faculty of Chemical and Biological Sciences, Islamia University of Bahawalpur , Bahawalpur , Punjab , Pakistan
3 Department of Soil Science, Faculty of Agriculture and Environment, Islamia University of Bahawalpur , Bahawalpur , Punjab , Pakistan
4 Institute of Forest Sciences, Faculty of Agriculture and Environment, Islamia University of Bahawalpur , Punjab , Pakistan
Hussein Mona
Electronic publication date: 2024 Feb 29
Publication date: 2024
Volume: 12
Electronic Location ID: e16782
Received 2023 Aug 2; Accepted 2023 Dec 19
Copyright: ©2024 Iqbal et al.
Copyright year: 2024
Copyright holder: Iqbal et al.
License: This is an open access article distributed under the terms of the Creative Commons Attribution License, which permits unrestricted use, distribution, reproduction and adaptation in any medium and for any purpose provided that it is properly attributed. For attribution, the original author(s), title, publication source (PeerJ) and either DOI or URL of the article must be cited.
License URL: https://creativecommons.org/licenses/by/4.0/

Keywords: Consumer packaging, Polyethylene, Polypropylene, Polyvinylchloride, Food safety, Household pests, Integrated pest management, Non-chemical control, Entomology, Agriculture

Funding: Higher Education Commission of Pakistan PM-IPFP/HRD/HEC/2012/4023 This research work is related to a research project funded by Higher Education Commission of Pakistan (PM-IPFP/HRD/HEC/2012/4023) in favour of Muhammad Waqar Hassan. The funders had no role in study design, data collection and analysis, decision to publish, or preparation of the manuscript.

==============================
Ants belonging to the Monomorium indicum (Formicidae: Hymenoptera) species are ubiquitous insects that are commonly associated with household settings in Pakistan. Packaged foodstuffs are easily destroyed by household ants when packaging is made with materials that have a high susceptibility. This study evaluated the susceptibility of three common flexible plastic packaging materials namely: opaque polyethylene, transparent polyethylene and polypropylene, which were each tested at thicknesses of 0.02 mm for their susceptibility against M. indicum. Except opaque polyethylene which is only available at 0.02 mm thickness, both transparent polyethylene and polypropylene were tested at higher thickness of 0.04 mm and 0.06 mm also against M. indicum. In order to simulate household settings, experiments were conducted at the faculty building of the agriculture and environment department of The Islamia University of Bahawalpur, Pakistan during summer vacations when the building was quiet. Different corners were selected near water sources for maximum exposure to the largest number of ants. Experimental cages used for the experiment were built with wood and 2 mm iron gauze to allow only ants to enter the cages. Daily activity of ants was used as an infestation source in cages. Experiments were run over three time spans of fifteen days each from June 20th 2022 to August 15th 2022. Results showed all packaging materials were susceptible against M. indicum at the 0.02 mm thickness level. Polypropylene was susceptible at 0.04 mm thickness but resistant to ants at 0.06 mm thickness, whereas polyethylene was still susceptible to ants at the higher thickness of 0.06 mm. Correlation of packaging damage with weather factors showed that temperature had a positive relationship, while relative humidity had a negative association with M. indicum attack. Overall correlation of packaging damage with packaging thickness showed packaging thickness was negatively associated with packaging damage from the ants. Because major cutting role is performed by the mandibles, we studied mandibles of ants and three frequent pests of packaged foodstuff namely Rhyzopertha dominica, Tribolium castaneum and Trogoderma granarium. The results showed that ants had the largest mandible and frontal mandibular tooth lengths compared with the mandibles and frontal teeth of the common stored product pests, indicating M. indicum household ants have a higher pest status for packaged foodstuffs compared to common stored product pests. Although the thickness of the flexible plastic packaging was a major factor against household ants, the study results recommend the use of polypropylene with a thickness of at least 0.06 mm as foodstuff packaging against household ants compared with polyethylene packaging, which was found to be susceptible to ants even at 0.06 mm thickness.

Introduction

Ants are a common, highly-observable group of insects and more than 12,000 species of ants have been identified worldwide (Hammond, 2011; Malik, Arshad & Jamil, 2013). Ants can thrive in many types of environments and account for 15–25% of all living land animals (Schultz, 2000). They are one of the most common household pests. Ants are also social insects that live in colonies. Workers scavenge food and brings it to a central nest, which is frequently built far from the food source (Beatson Campbell, 1991). Ants are common anywhere they can obtain food and water (Li et al., 2005). Household ants can infect utensils and food, causing illness when these infected items are consumed by people (Garcia et al., 2011). The environment of Pakistan is conducive to the survival and growth of ants (Máximo et al., 2014).

The widespread use of vulnerable packaging materials for food goods has caused significant problems since losses from pest infestations of packaged foods equate to the total cost of cultivating, harvesting, transporting, preparing and packing the food (Mullen & Mowery, 2000). Any exhaustive examination of pest control in the food sector must consider food packaging in the eradication or prevention of insect infestation. Many companies have implemented package-testing programs to improve the resistance of packaging to insect attack (Mullen & Mowery, 2000), and insect-resistant packaging is the most frequent method of insect infestation prevention aside from insecticides or repellents (Mullen & Highland, 1988). Frequent causes of insect infestation of food include transportation-related issues or lengthy storage in suboptimal conditions at a warehouse or on a supermarket shelf.

Insect-resistant packaging is an effective method for preventing insect-related damages to packaged foodstuffs. Foodstuff packaging derived from plastics like polypropylene with a thickness of 0.04 mm was found to be resistant to insect penetrations or invasions of a major stored grain borer, Rhyzopertha dominica (Hassan et al., 2016). Previous research done by the study researchers evaluating the susceptibility of flexible plastic foodstuff packaging found that major stored grain insects were able to tear plastic packaging and consume or remove some of the food inside, leading to weight loss in the packaged foodstuffs (Qasim et al., 2013; Hassan et al., 2014; Hassan et al., 2016; Yar et al., 2017; Akram et al., 2018; Hussain et al., 2019; Waheed et al., 2022). However, little or no research is currently available on the susceptibility of commonly-used flexible plastic foodstuff packaging against household ants.

Common observations in household settings indicate that ants can be more threatening to foodstuffs packaging because of their ability to reach stored food materials through the smallest possible openings. In Pakistan, ants are usually controlled in homes through insecticidal powders sprinkled along the sighted trails of ants and their observed places of origin. Though these insecticides claim to be totally safe for indoor use, the use of chemical insecticides in residential places is riskier than in outdoor fields. Toxicity classifications and safety classifications between pesticides being applied in field crops and those manufactured for household use should differ based on indoor or outdoor use and human proximity. Although a number of social insect pests like cockroach and ants have been effectively managed using baits (Bennett et al., 2013), many bait-based initiatives have failed against household ants because of pesticide resistance and insufficient level of attractiveness (Rust, Reierson & Klotz, 2002; Krushelnycky & Rosemary, 2008) and baits containing insecticides are also not without danger when used in human residences.

Insect-resistant packaging of food material is the last line of defense for food producers against insect attack (Hou, Fields & Taylor, 2004) Testing different types of packaging and thickness levels against household ants is necessary because different insect pests like Tribolium castaneum, Trogoderma granarium and R. dominica have significantly different abilities in chewing substrate materials (Hassan et al., 2021). Stored product pests vary in their ability to penetrate packages (Arthur & Phillips, 2003). Therefore, the current study was designed to evaluate three commonly-used flexible plastic foodstuff packaging types for susceptibility against household ants: transparent polyethylene (low density), opaque polyethylene (high density) and polypropylene. Small plastic pouches were filled with fruit cake and placed in natural ant foraging locations in a household setting.

Material and Methods

Study location and ant specimens

This study was conducted at the faculty building of the agriculture and environment department on the Baghdad campus of The Islamia University of Bahawalpur, Pakistan. The natural movement of ants was used as an infestation source in experiments. Experiments were performed during summer vacations, when academic activities were limited and the academic building was usually quiet. Ants were abundant at the experimental locations within the agriculture and environment faculty building. Ant specimens were collected with moist camel hair brush, preserved in 70% ethanol and subsequently identified as Monomorium indicum Forel at the Insect Biodiversity Laboratory, Department of Entomology at The Islamia University of Bahawalpur.

Experiment cages

Cages were built for this experiment with wood and two mm iron gauze, with each cage 8 × 8 square inches in size. In total there were nine such cages to retain three replicates for testing each packaging thickness. Cages were built so that ants could enter the box, but no other damaging pests (rodents, lizard, cats and squirrels) could enter.

Packaging materials

The most common flexible plastic foodstuff packaging types being used in Pakistan include opaque polyethylene (high density), transparent polyethylene (low density) and polypropylene. For this study, plastic materials were purchased from a wholesale plastic market in Lahore at rate of 400 rupees per kg. Mean thickness of the different packaging materials was identified using a digital micrometer (Mitutoyo Corporation, Kawasaki, Japan). All three types of materials were available in 0.02 mm thicknesses. Transparent polyethylene and polypropylene were also available in 0.04 mm and 0.06 mm thickness levels, but not high density or opaque polyethylene. These plastic packaging films were purchased and used in the experiments to evaluate their susceptibility against M. indicum household ants. Small bags of these plastic films (8 × 10 cm) were prepared in the laboratory using a pair of scissors and an impulse (heat) sealer.

Packaged food

Fresh fruit cake, purchased from the local market, was used inside the packaging to check the susceptibility of the packaging types against household ants. Prepared bags of each type and thickness of packaging were filled with one 18 g fresh fruit cake slice, which was weighed on an electric scale. After adding the fruit cake slices, the plastic bags were sealed with a heat sealer.

Experimental setup

Three types of 0.02 mm thick plastic bags—opaque polyethylene (high density), transparent polyethylene (low density) and polypropylene—were filled with fruit cake and sealed with a heat sealer. There were no prior vents or holes in the bags that would allow insects to enter. These packages were then placed inside a cage. Three similar packaging types but without fruit cake (control treatments) were also placed in the experimental cage. The cage was then closed and locked to restrict entry of any foreign objects. Two other cages were prepared in the same manner to maintain three replicates for 0.02 mm thick packaging. Similar methods were used to evaluate 0.04 mm and 0.06 mm thickness packaging, with three replicate cages prepared with 0.04 mm thick transparent polyethylene and polypropylene plastic bags, both with and without food, and three replicate cages prepared with 0.06 thick transparent polyethylene and polypropylene plastic bags, both with and without food.

One cage for each thickness of packaging was placed at each of three different locations near a water source where ants had previously been observed in the faculty building of the agriculture and environment department. Each of the three locations had three cages, one for each thickness level of packaging, for a total of nine cages. This experiment was under observation for the whole study period to reduce any possible outside disturbances. The cages were visited daily and the number of holes in the packaging data was collected after every five days for fifteen days for this experimental setup. This first experiment lasted from 20th June to 5th July 2022.

Every five days, the cages were opened, the bags were removed, and damage to the packaging, measured by number of holes, was observed. Packages displaying any sealing defects were immediately replaced with the same type of packaging to avoid ant invasions through holes not created by the ants (Mullen, Vardeman & Bagwell, 2012). The number of holes in the packaging was measured and then the bags were opened to measure weight loss of the fruit cake caused by ants, using the following formula:

% age weight loss=Initial weight−Final weightInitial weight×100.

This experimental setup was maintained for fifteen days and data was gathered for each of the three thicknesses. Following the first experiment, the same experimental setup was repeated from 10th July to 25th July 2022 (second experiment) and finally from 1st August to 15th August 2022 (third experiment) (Table 1). For the second and third experiments, new packages were used along with newly-packaged fruit cake (18 g) for each packaging type and thickness.

Table 1 Experimental setups and weather data of respective dates.

Experimental setups	Data recorded	Temperature °C	Relative humidity %	
20.06.2022	25.06.2022	40	43	
30.06.2022	38	51	
05.07.2022	40	55	
10.07.2022	15.07.2022	35	65	
20.07.2022	38	58	
25.07.2022	31	91.5	
30.07.2022	04.08.2022	37	65	
09.8.2022	38	59.75	
14-08-2022	33	73.5	

Study of insect mandibles

Mandibles of M. indicum and three common pests of stored products, Rhyzopertha dominica, Tribolium castaneum, and Trogoderma granarium were all studied under microscope. Three specimens of each insect type were selected. The head region of each specimen was separated using fine forceps and surgical blade no. 14 and then mounted on a clean glass slide in glycerin (50%). Mandibles were oriented under a camera (Model HD 1500 T, Meiji, TECHNO, Saitama, Japan) fitted with a trinocular stereoscope microscope (Labomed, CXR3, Labo America, Inc., Fremont, CA, USA) with installed software (T Capture Version 3.9 digital software; T Capture, 2017) on a laptop computer (Dell Core i3, 10th Gen). The mandibles of the insect specimens were orientated for proper measurements and visual comparisons, photo captured and saved with proper labelling for future reference. The images were opened with T Capture software, which was calibrated using the micrometer scale (1 mm). The mandibles, as well as mandibular frontal tooth of three specimens for each insect type, were measured. The images, along with the measurements, were saved and the respective values were tabulated in Microsoft Excel 2021 (Version 2019; Microsoft, Bellevue, WA, USA) for further data analyses.

Data analysis

Data was statistically analyzed using SPSS software (version 2016; SPSS Inc., 2007). Data was analyzed separately for each thickness level using one-way ANOVA in which different packaging types, both with and without food, served as independent variables to see the effect of packaging types of each thickness level on number of holes and percent weight loss of packaged fruit cake, with number of holes and percent weight loss serving as dependent variables. One-way ANOVA was also performed to see the effect of the three experiment dates on number of holes and weight loss of packaged fruit cake, with experiment dates serving as the independent variable and number of holes and percent weight loss of the fruit cake serving as dependent variables. Mean values were separated post hoc at a 5% level of probability using a Tukey HSD test. For each thickness level, correlation (Pearson) was also tested between number of holes created by M. indicum (damage) and weather data, including temperature and relative humidity, of the three experiment dates. To see the overall effect of packaging thickness on damage, a Pearson correlation analysis was performed on all the data between number holes in all thickness levels (omitting high density polyethylene in 0.02 thickness to standardize data along three thickness levels) and packaging thickness. Finally, the measured mandible of M. indicum was statistically compared with the measured mandibles ofthe three frequent pests of packaged foodstuff: Rhyzopertha dominica, Tribolium castaneum and Trogoderma granarium. An analysis of variance one-way ANOVA was also performed in which lengths of the mandible and frontal tooth of M. indicum and three storage pests served as dependent variables, and insect types served as independent variables. Means values were separated post hoc by a Tukey HSD test at a 5% level of probability.

Results

Effect of packaging type on number of holes in packaging and weight loss of packaged fruit cake caused by M. indicum household ants

Figure 1 shows the effect of packaging types with 0.02 mm thickness on damage (holes in packaging and percent weight loss of fruit cake) caused by M. indicum household ants. The results showed ants created the highest number of holes (2.00) in polyethylene high density packaging, followed by low density polyethylene bags (1.56), with the least number of holes found in polypropylene (1.22) packaging. Damage was not observed in any of the packaging types without food material (F5, 53:1.832; P: .124).

Figure 1 Effect of packaging types with 0.02 mm thickness on holes in packaging and weight loss in packaged fruit cake caused by M. indicum.

1, Packaging with food material; 2, Packaging without food material. Means comparison by Tukey HSD test at a 0.05 level. Small letters are along error bars for number of holes and capital letters along percent weight loss bars. Different letters along same bars show significant differences in means.

Weight loss of packaged fruit cake was recorded in packaging where holes were created by ants. Percent weight loss due to ants feeding was highest in the polyethylene high density (39.64%) packaging, followed by polypropylene (21.03%) packaging, with the lowest weight loss percentage observed in polyethylene low density (18.56%) packaging. Zero weight loss was recorded in packaging without holes (F5, 53: 2.762; P: .028).

In 0.04 mm thick packaging, the highest average number of holes was recorded (0.11) in polypropylene packaging with fruit cake, while no holes occurred in polyethylene packaging and packaging without fruit cake (F3, 35: 1.000; P: .405). Similarly, percent weight loss was only recorded in polypropylene packaging (8.09%) with fruit cake, but no weight loss was recorded in packaging without holes (F3, 35: 1.000; P: .405; Fig. 2).

Figure 2 Effect of packaging types with 0.04 mm thickness on holes in packaging and weight loss in packaged fruit cake caused by M. indicum.

1, Packaging with food material; 2, Packaging without food material. Means comparison by Tukey HSD test at a 0.05 level. Small letters are along error bars for number of holes and capital letters along percent weight loss bars.

In 0.06 mm thick packaging, the highest average number of holes was recorded (0.44) in polyethylene packaging with fruit cake, while no holes occurred in polypropylene packaging and packaging without fruit cake. Similarly, percent weight loss was only recorded in polyethylene packaging (5.36%) with fruit cake, but no weight loss was recorded in packaging without holes (F3, 35: 1.000; P: .405; Fig. 3).

Figure 3 Effect of packaging types with 0.06 mm thickness on holes in packaging and weight loss in packaged fruit cake caused by M. indicum.

1, Packaging with food material; 2, Packaging without food material. Means comparison by Tukey HSD test at a 0.05 level. Small letters are along error bars for number of holes and capital letters along percent weight loss bars.

Effect of experiment dates on number of holes in packaging and weight loss of packaged fruit cake caused by M. indicum household ants

At a packaging thickness level of 0.02 mm, ants were able to cause damage on all three dates of experiments ranging from 25th June to 5th July, 15th July to 25th July and from 5th August to 15th August during 2022 (Fig. 4). In these date ranges, the highest number of holes was recorded (1.28) during the first experiment, followed by the second experiment (1.06), with the lowest number of holes recorded (.06) in the third experiment (F2, 53: 1.806; P: .175). The highest weight loss percentage of packaged fruit cake was recorded in the first experiment (23.72%), followed by the second experiment (11.78%), with the lowest weight loss percentage (4.11%) recorded in the third experiment (F2, 53: 1.818; P: .173).

Figure 4 Effect of experiment dates on holes in packaging and weight loss in packaged fruit cake in 0.02 mm thick packaging caused by M. indicum.

1, First experiment (20th June to 5th July 2022); 2, second experiment (10th July to 25th July); 3, third experiment (1st August to 15th August). Means comparison by Tukey HSD test at a 0.05 level. Small letters are along error bars for number of holes and capital letters along percent weight loss bars.

At a packaging thickness level of 0.04 mm, 0.08 holes created by M. indicum were recorded in the first experiment, however no holes were created in 0.04 mm thick packaging in the second or third experiments. Accordingly, 6.06% weight loss of packaged fruit cake was recorded during the first experiment, but no weight loss was recorded in the second or third experiments (F2, 35: 1.000; P: .379; Fig. 5).

Figure 5 Effect of experiment dates on holes in packaging and weight loss in packaged fruit cake in 0.04 mm thick packaging caused by M. indicum.

1, First experiment (20th June to 5th July 2022); 2, second experiment (10th July to 25th July); 3, third experiment (1st August to 15th August). Means comparison by Tukey HSD test at a 0.05 level. Small letters along error bars for number of holes and capital letters along percent weight loss bars.

At a packaging thickness level of 0.06, 0.33 holes created by M. indicum were recorded in the first experiment, however no holes were created in 0.06 mm thick packaging in the second or third experiments. Similarly, 5.36% weight loss of packaged fruit cake was recorded in the first experiment, but no weight loss was recorded in the second or third experiments (F2, 35: 1.000; P: .379; Fig. 6).

Figure 6 Effect of experiment dates on holes in packaging and weight loss in packaged fruit cake in 0.06 mm thick packaging caused by M. indicum.

1, First experiment (20th June to 5th July 2022); 2, second experiment (10th July to 25th July); 3, third experiment (1st August to 15th August). Means comparison by Tukey HSD test at a 0.05 level. Small letters are along error bars for number of holes and capital letters along percent weight loss bars.

Correlation of damage caused by M. indicum with weather factors and packaging thickness

The correlation analysis results showed that in all three packaging thickness levels, temperature had a strong positive relationship with damage to packaging caused by M. indicum, while relative humidity had a strong negative effect on packaging damage. A correlation analysis of overall data for all thickness levels showed packaging thickness had a negative correlation with damage caused by M. indicum (Table 2).

Table 2 Correlation of holes and percent weight loss in packaging with weather factors and packaging thickness.

	Correlation of weather factors and packaging material thickness with damages	
Thickness	0.02 mm	0.04 mm	0.06 mm	
Factors	r	P	r	P	R	P	
Temperature	0.5083	0.6606	0.9066	0.2774	0.9066	0.2774	
Relative humidity	−0.6003	0.5901	−0.9476	0.207	−0.9476	0.207	
Thickness effect	r−0.2662; P: 0.0517	

Study of M. indicum mandibles in relation to mandibles of major stored grain insect pests

Figure 7 shows a comparison of mandibular length and frontal tooth length between M. indicum household ants and three major stored product pests. The results showed M. indicum had significantly longer mandibles compared with the mandibles of three common stored product pests (F3, 11: 94.551; P: < 0.001). The highest recorded mean length of mandible was 400.67 µm for M. indicum, followed by 241.67 µm for adult R. dominica, and 201.33 µm for T. castaneum, and the lowest recorded mean length of mandible was 174.33 µm for T. granarium larva. Mandibular frontal tooth length was highest in M. indicum (124.00 µm), followed by both R. dominica and T. castaneum at 81 µm, and the lowest length was 32.00 µm in T. granarium larva (F3, 11: 68.601; P: < 0.001).

Figure 7 Mean mandibular and mandibular largest tooth lengths of M. indicum and three common stored product pests.

Means comparison by Tukey HSD test at a 0.05 level. Small letters are along error bars for mandibular lengths and capital letters along mandibular large tooth length bars. Different letters along same bars show significant differences in means.

Discussion

This study was conducted to test the susceptibility of commonly-used flexible plastic foodstuff packaging films against M. indicum household ants. Small bags of different types of packaging films at 0.02 mm, 0.04 mm and 0.06 mm thicknesses were created and tested for their susceptibility against the naturally-foraging household ants. The results showed at 0.02 mm thickness, the highest susceptibility was in high density opaque polyethylene films, followed by low density transparent polyethylene and then polypropylene. Weight loss of the fruit cake was also significantly higher in opaque polyethylene followed by polypropylene and then transparent polyethylene.

No holes were observed in any type of packaging without fruit cake. This may be because ants could distinguish between packaging with and without fruit cake due to odours emitting from the fruit cake bags. Barrier materials have been reported to prevent food odors from escaping the package, resulting in a package that is invisible to invading insects (Sacharow & Brody, 1987). Mullen, Vardeman & Bagwell (2012) also emphasized the importance of odor barriers to prevent insect infestation in packaged foodstuffs (Mullen, Vardeman & Bagwell, 2012).

All packaging materials tested at 0.02 mm thickness in this study were susceptible to foraging ants, which were able to create holes and eat the fruit cake. These results are in line with the findings of Chung et al. (2011) that plastic packaging thickness is an important factor on insect damage to food. They also observed more penetrations by insects in thinner packaging. Similar results have also been recorded about thickness effect on penetration by larvae (Li et al., 2014).

In packaging with 0.04 mm thickness, less damage, as measured by number of holes in the packaging and weight loss of packaged fruit cake, occurred due to ants. At this thickness level, damage was only recorded in polypropylene packaging. No damage was recorded in polyethylene packaging or in packaging without fruit cake.

In 0.06 mm thick packaging, damage was only recorded in polyethylene packaging. Thicker packaging might mean less food odor emission through packaging films and the thicker packaging may have been harder for ants to penetrate. These results are in agreement with earlier reports which stated that when packaging was used with extra cover these were resistant to insect penetration than when used alone (Mullen & Mowery, 2000). Therefore, packaging thickness is major factor in resistance against household ants.

Compared with polypropylene, polyethylene proved more susceptible, with damage observed at a higher thickness level of 0.06 mm. Marouf & Momen (2007) in a comparative study between polyethylene, polypropylene and polyvinylchloride packaging, found polypropylene with comparatively less thickness as an ideal liner of bags to resist insect penetrations.

The results of this study are also in line with earlier study results of these researchers, which showed that polypropylene packaging proved resistant to damage caused by insects like punctures, holes and penetrations, compared with polyethylene (Hassan et al., 2016). According to Pacheco & Wiendl (1989), polypropylene is an effective wrapper for packaged beans to stave off common bean weevil penetration.

Although there are many factors known to affect an insect’s ability to tear packaging, one of them is the surface or texture of the packaging films. The smooth surfaces of plastic bags are known to affect insect walking (Domingue et al., 2022) and smooth surface texture might be one of the reasons explaining polypropylene packaging resistance against insect penetration. Polypropylene has been reported to have a more slippery surface than polyethylene (Cline, 1978; Jassim, Mubark & Falih, 2022).

An analysis of the effect of experiment dates showed higher levels of ant damage in 0.02 mm thick packaging in the first experiment during late June to early July than in later dates. In tests of thicker packaging levels of 0.04 mm and 0.06 mm, ant damage was only recorded in the first experiment, with no damage recorded in the later experiments. The correlation analysis of weather factors and thickness level study showed temperature had a positive relationship with ant damage, and relative humidity had a negative relationship with ant damage. These results showed ant infestations were more common in hotter and drier periods of the season. These results are in agreement with the study findings of Barbani (2003), which showed a similar relationship between the foraging activity of ants and weather factors.

The results of this study showed ants are more harmful to foodstuff packaging than the majority of stored grain pests against a packaging thickness of 0.04 mm, which proved resistant to these pests. To confirm this finding, the mandibles of ant species M. indicum were compared with the mandibles of three common stored product pests: R. dominica, T. castaneum and T. granarium. The microphotography of the mandibles showed that the mandibles of M. indicum were significantly larger than those of the three common stored product pests. Frontal tooth length was also highest in ants, confirming that M. indicum is more hazardous to foodstuff packaging than common stored grain pests. Polypropylene packaging is recommended for foodstuffs at a thickness of at least 0.06 mm as a protection against household ants.

Supplemental Information

Figure S1 Cages for packaging evaluation against M. indicum

Figure S2 Measured mandible of M. indicam, LMT, Large mandibular tooth

Figure S3 Measured mandible of R. dominica, LMT, Large mandibular tooth

Figure S4 Measured mandible of T. castaenum, LMT: Large mandibular tooth

Figure S5 Measured mandible of T. granarium larva, LMT, Large mandibular tooth

Data S1 Damages observed in packaging types separately for different thickness levels caused by M. indicum and measured lengths of mandibles and large mandibular teeth for M. indicum and three common stored product pests

Additional Information and Declarations

Competing Interests

Author Contributions

Data Availability

The authors declare there are no competing interests.

Muhammad Junaid Iqbal conceived and designed the experiments, performed the experiments, prepared figures and/or tables, authored or reviewed drafts of the article, and approved the final draft.

Muhammad Waqar Hassan conceived and designed the experiments, performed the experiments, analyzed the data, prepared figures and/or tables, authored or reviewed drafts of the article, and approved the final draft.

Ghulam Sarwar conceived and designed the experiments, performed the experiments, prepared figures and/or tables, authored or reviewed drafts of the article, and approved the final draft.

Moazzam Jamil conceived and designed the experiments, authored or reviewed drafts of the article, and approved the final draft.

Tanveer Hussain conceived and designed the experiments, authored or reviewed drafts of the article, resources management, and approved the final draft.

The following information was supplied regarding data availability:

The raw data is available in the Supplemental File.

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
