# Peer review of "Susceptibility of flexible plastic foodstuffs packaging against Monomorium indicum (Hymenoptera: Formicidae) household ants"

_PeerJ, doi:10.7717/peerj.16782_

## Round 0.1 · original submission · Major Revisions

Dear authors,

Your manuscript is a good study and your results about packaging susceptibility showed that ants are more harmful to foodstuff packaging than the majority of stored grain pests. This article needs language editing in all its parts. The reviewer comments should be considered. It would be better for this article to be revised by a fluent English editor before re-submission.

**Language Note:** The Academic Editor has identified that the English language must be improved. PeerJ can provide language editing services - please contact us at copyediting@peerj.com for pricing (be sure to provide your manuscript number and title). Alternatively, you should make your own arrangements to improve the language quality and provide details in your response letter. – PeerJ Staff

·

Basic reporting

Peer review report on manuscript Ref: Submission ID peerj-reviewing-88767-v0
Susceptibility of flexible plastic packaging for foodstuffs against the household ants Monomorium indicum Forel (Hymenoptera: Formicidae)
Original submission
Recommendation: major revision

Overview and general recommendations
It is a good study. But there are some points need to correct in review file

Experimental design

good

Validity of the findings

original

Reviewer 2 ·

Basic reporting

.

Experimental design

.

Validity of the findings

.

Additional comments

Title of Manuscript: Susceptibility of flexible plastic packaging for foodstuff against the household ants Monomorium indicum Forel (Hymenoptera: Formicidae)


This manuscript needs revision before it is published. The language, syntax and flow of information need to be addressed. It would be more appropriate if it is reviewed to improve the language before submission.

Some review comments are below:

Ln 33-37: Briefly explain how ants accessed the material before defining results.

Ln 44: You can list here what stored pests were used for mandibular comparisons.

Ln 36-51: Were both opaque polyethylene and transparent polyethylene susceptible at 0.06 mm thickness? This indicates that both materials were also susceptible at 0.04 mm thickness, which only leaves polypropylene resistant at 0.06 mm thickness. Be consistent in using singular or plural for both the mandibles and the teeth. Define for the reader why you are associating length of mandibles and mandibular teeth.

Ln 55: Specify if this is the number of worldwide species or regional.

Ln 55-61: I would restructure this paragraph. Numerous sentences may be combined to avoid repeated use of pronouns for ants.

Ln 58: Do you mean worker caste?

Ln 60: Remove semicolon and restructure sentence.

Ln 75: You can name the major stored grain borer for the instant of knowledge of the readers.

Ln 80-82: Rephrase to improve the flow. I would not use “due to” and “because of” in a single sentence here.

Ln 90: Pesticide resistance and insufficient appeal – against what? Mentioning of insect or other pest will be more appropriate than just giving reference to ease the understanding of the readers.


Ln 96: Same comment as or Ln 90 above.

Ln 98 -103: Confusing statements and syntax.

You have somewhat adequate literature referenced but in almost all cases the reader has to consult the reference to understand what organism(s) you are referring to.
Ln 117: How ants were collected and in what % ethanol they were preserved?

Ln 123: Use of “was” and “is” confusing.

Ln 121-125: Need to improve flow and syntax.

Ln 199-210: You may list the other insects used for mandibular comparisons. These are confusing statements. Sentences may be articulated better to improve understanding.


Statistical design is ok but Results and Discussion sections also need to be promptly revised to correct grammar and syntax mistakes. Figure legends need revisions.

---

## Round 0.2 · accepted · Accept

Dear authors,

Thank you for addressing all reviewer's recommendations and comments. I went through the re-submitted version myself and I am pleased that you responded to all comments. The language and structure of the current manuscript are much better. Therefore, your manuscript is now ready for publication.